# Multi-Modal Synergistic 99mTc-TRODAT-1 SPECT and MRI for Evaluation of the Efficacy of Hyperbaric Oxygen Therapy in CO-Induced Delayed Parkinsonian and Non-Parkinsonian Syndromes

**DOI:** 10.3390/antiox11112289

**Published:** 2022-11-18

**Authors:** Skye Hsin-Hsien Yeh, Chuang-Hsin Chiu, Hung-Wen Kao, Ching-Po Lin, Yu-Hus Lai, Wen-Sheng Huang

**Affiliations:** 1National Defense Medical Center, School of Medicine, Taipei 114201, Taiwan; 2National Defense Medical Center, Department of Nuclear Medicine, Tri-Service General Hospital, Taipei 114201, Taiwan; 3Department of Medical Imaging, Hualien Tzu Chi Hospital, Buddhist Tzu Chi Medical Foundation, Hualien 970473, Taiwan; 4Department of Radiology, School of Medicine, Tzu Chi University, Hualien 97004, Taiwan; 5Institute of Neuroscience, National Chaio Tung University, Taipei 112304, Taiwan; 6Department of Neurology, Cheng-Hsin General Hospital, Taipei 112304, Taiwan; 7Department of Nuclear Medicine, Cheng-Hsin General Hospital, Taipei 112304, Taiwan

**Keywords:** carbon monoxide-induced parkinsonism, oxidative stress, Tc-99m TRODAT-1 SPECT, delayed neuropsychiatric syndromes, hyperbaric oxygen therapy

## Abstract

Background: Delayed neuropsychiatric syndrome (DNS) is characterized by motor dysfunction after acute carbon monoxide (CO) poisoning. We examined the relationship between dopamine transporter (DAT) loss using kit-based Tc-99m-TRODAT-1 (DAT single-photon emission-computed tomography (SPECT) radioligand) and globus pallidus necrosis on MRI, DAT availability before and after hyperbaric oxygen therapy (HBOT), and feasibility of Tc-99m-TRODAT-1 as an index for parkinsonian syndrome in CO poisoning. Methods: Twenty-one CO-intoxicated patients (mean ± SD age, 38.6 ± 11.4; range, 20–68 years) with DNS underwent Tc-99m-TRODAT-1 SPECT and MRI before HBOT and follow-up Tc-99m-TRODAT-1 SPECT to assess DAT recovery. Neurological examinations for Parkinsonism were performed after development of DNS. Results: Over 70% (15/21) of DNS patients showed globus pallidus necrosis on MRI. Significantly lower bilateral striatal DAT availability was associated with globus pallidus necrosis (*p* < 0.005). Moreover, 68.4% (13/19) of DNS subjects with Parkinsonian syndrome had lower bilateral striatal DAT availability vs. non-parkinsonian subjects pre- or post-HBOT. The SURs for both striata increased by ~11% post-HBOT in the Parkinsonian group; however, the left striatum presented a significantly higher DAT recovery rate than the right (*** *p* < 0.005). Conclusions: Coupled Tc-99m TRODAT-1 SPECT and MRI could assist evaluation of Parkinsonism risk and indicate DAT availability after HBOT in CO-poisoned patients with DNS.

## 1. Introduction

Carbon monoxide (CO) poisoning typically occurs on breathing in excessive levels of CO, a toxic gaseous by-product of incomplete combustion of carbon-based fuels and substances [1]. CO is the most common lethal poison worldwide and a leading cause of poisoning deaths in the United States and Europe [2,3]. CO has a 250-fold higher affinity for hemoglobin than O_2_. The lower delivery of O_2_ to the CNS in individuals exposed to CO leads to ventilator stimulation and higher CO uptake and HbCO levels, which causes respiratory alkalosis [4].

Human autopsies have demonstrated CO poisoning affects a number of regions in the brain including the globus pallidus, cerebral cortex, hippocampus, striatum, and caudate putamen [5]. Delayed neuropsychological sequelae (DNS) are the most frequent form of morbidity after CO poisoning can begin as early as a few days and weeks after exposure and may last up to one year or longer [6,7,8]. Up to 40% of survivors of CO poisoning may develop DNS within weeks after an initial complete clinical recovery from acute poisoning [9]. More than of 40% CO-poisoned patients develop non-Parkinsonism syndrome such as depression, anxiety, and cognitive dysfunction [10], and Parkinsonism has been reported to occur in 9.5% of CO-poisoned patients [11,12]. The signs of Parkinsonism include a masked face, hypokinesia, a short-step gait, higher muscle tone (rigidity), the grasp reflex, the glabella sign, and retropulsion [12,13].

CO may cause widespread damage in the brain; however, the corpus striatum is especially vulnerable [14]. Striatal damage could explain the Parkinsonian signs that can develop after severe CO poisoning. The pathogenesis of CO-induced Parkinsonism is poorly characterized, though the CO-associated reduction in oxygen to the tissues and hypoxia reduce the rate of turnover of dopamine. An excess of dopamine and its metabolites may lead to neurotoxicity in acute CO poisoning [10].

Moreover, CO directly induces a variety of cellular changes and oxidative stress-neurodamage via a number of mechanisms [15]. CO-induced inhibition of mitochondria reduces oxidative phosphorylation and production of ATP and increases the tissue and brain levels of superoxide radicals, which subsequently cause further mitochondrial dysfunction and neuronal damage in Parkinsonism syndrome [16].

Inflammation also contributes to DNS after CO poisoning [17,18] as CO activates degranulation of neutrophils, which leads to release of myeloperoxidase (MPO). In turn, MPO amplifies inflammation by further activating neutrophils, which trigger downstream pathways that generate reactive oxygen species (ROS). High levels of ROS lead to lipid peroxidation, form adducts with myelin basic protein (MBP), and attract lymphocytes and activate microglia [16,19].

Overall, hypoxia, CO-associated dopaminergic neurotoxicity, in Parkinsonism is associated with oxidative stress and neuroinflammation [20].

Furthermore, motor symptoms related to motor/action control are the major signs Parkinsonism [21]. Parkinsonism also occurs in approximately 20–30% of patients with frontotemporal lobar degeneration (FTLD) [22]. Loss of frontotemporal neurons reduces dopaminergic input, which leads to less positive motor activity and increases negative motor inhibition [13]. The basis of the motor control abnormalities (i.e., movement slowing, Bradykinesia, rigidity/higher muscular tone) in Parkinsonism remain unclear and are complicated by the incomplete understanding of normal motor control (i.e., movement speed and normal muscle tone) [21]. Battaglia et al. (2022) studied vicarious fear learning among healthy participants and showed, for example, an image of severe acute respiratory syndrome coronavirus 2 (SARS-CoV-2) promotes action control abilities [23] and emotional stimuli may increase sensory representation and/or attentional processing and thus improve inhibitory performance [24]. This evidence supports the fact that patients with Parkinsonism taking dopamine agonists develop impulse control disorders; dopamine has been suggested to mediate response inhibition due to its effects on the basal ganglia [25].

Neuroimaging plays an important role in the study of this entity in both the acute phase, to confirm the clinical diagnosis and provide prompt treatment, and in the chronic phase, to evaluate possible neurological sequelae [26]. Brain magnetic resonance imaging (MRI), as well as neuropsychological testing, are useful for the diagnosis and assessment of the severity of CO toxicity. Neuroanatomical abnormalities following CO poisoning are detected on MRI in the globus pallidus, and substantia nigra [27,28]. A functional nuclear imaging study using SPECT 99mTc-TRODAT-1, a specific imaging agent for dopamine transporter (DAT), reported reduced DAT availability during a 6-month follow-up period was associated with impaired cognitive performance, which indicates a crucial role for DAT in the recovery of executive function following CO poisoning [29]. Measurement of DAT by 99mTc-TRODAT-1 SPECT has clinical significance and is highly related to neurobehavioral scores and the severity of parkinsonian syndrome in patients with CO intoxication [30].

Clinical evidence suggests hyperbaric oxygen therapy (HBOT) should be administered to all patients with CO intoxication to prevent the development of DNS [31,32]. Huang et al. (2017) suggested that HBOT is associated with lower short- and long-term mortality in patients with significant CO poisoning, especially those with acute respiratory failure and patients younger than 20 years of age [33]. However, the results of randomized comparative tests (RCTs) and subsequent reports have been conflicting [34,35]. Therefore, it is unclear whether HBOT therapy during the acute phase of CO poisoning prevents neurological sequelae. Based on MR imaging, Lo et. al. (2007) reported HBOT improved CO-induced demyelination in white matter [27], whereas Liu et al. found HBOT improved the lesions within bilateral cerebral white matter and bilateral or unilateral globus pallidus and periventricular white matter associated with cognitive behavior and psychiatric symptoms [36]. Although the importance of dopaminergic dysfunction and parkinsonian symptoms have been emphasized, to the best of our knowledge, there are no reports on the alterations in DAT imaging before and after HBOT in patients with DNS.

As described above, up to a third of survivors of CO-poisoned develop DNS and HBOT is thought to prevent these symptoms. However, the changes in DAT availability after HBOT treatment and the differences in DAT availability between patients with parkinsonism and non-parkinsonism DNS are unclear. Thus, a coupled imaging procedure based on structural MRI and functional 99mTc-TRODAT-1 was conducted using a modified version of the dopaminergic imaging in Parkinsonian syndromes 1.0 published by the European Association of Nuclear Medicine guidelines [37]. The aims of this study were (1) to evaluate 99mTC-TRODAT-1 SPECT to assess the severity of DAT damage associated with MRI findings after CO poisoning; (2) to assess DAT availability before and after HBOT using 99mTC-TRODAT-1 SPECT; and (3) to evaluate multi-modal synergistic 99mTC-TRODAT-1 SPECT and MRI as an index for parkinsonian or non-parkinsonian syndromes in patients with CO poisoning.

## 2. Materials and Methods

### 2.1. Patient Enrollment

This study was approved by the ethical committees and review board of Tri-Service General Hospital, Taipei, Taiwan (Institutional Review Board [IRB] #095-05-0024). A total of 21 subjects with suicidal attempts (14 men, 7 women; mean ± SD age, 38.6 ± 11.4 years; age range, 20–68) were recruited to this study. Patients were diagnosed with CO poisoning if they had an initial carboxyhemoglobin (COHb) level of over 10% or if carbon monoxide exposure and obvious CO-poisoning-related symptoms were confirmed. All patients received toxicology screening to rule out other poisoning etiologies and had not taken any medications for at least 3 months.

Patients were excluded from this analysis if they were <18 or >65-years-old or had a current or past history of neuropsychiatric disorders or a family history of movement disorders based on electronic health records from the Bureau of National Health Insurance (Taipei, Taiwan), as determined by a screening interview.

All subjects were initially comatose and regained full consciousness within hours to days after acute CO exposure. All subjects developed progressive neuropsychiatric impairments and delayed neuropsychiatric syndrome (DNS), such as slow responses, disorientation, apathy, amnesia, and parkinsonian features, after an apparent lucid interval of 30–60 days.

### 2.2. Study Design

All subjects underwent a 99mTC-TRODAT-1 brain SPECT and brain MRI to evaluate the availability of striatal dopamine transporter and presence of necrosis in the globus pallidus, respectively, <1 month after CO exposure (Table 1), before or during the initial section of hyperbaric oxygen therapy (HBOT) therapy. Follow-up TC-99m TRODAT-1 brain SPECT and MRI (84.6 ± 32.3 days from insult) was performed at 3 months after HBOT. Table 1 summarizes the clinical data for all subjects.

### 2.3. Hyperbaric Oxygen Therapy

Patients underwent HBOT at intervals of <1 week between CO-poisoning and the development of DNS. All patients received 100% high-flow oxygen treatment. HBOT was given under 100% oxygen at 2.5 atmospheres absolute (ATA) for 120 min in the initial and each session. Since the levels of CO poisoning and clinical severity varied substantially, the patients received different numbers of sessions of HBOT (range, 3–30) [27].

### 2.4. Imaging Studies

#### 2.4.1. TC-99m TRODAT-1 Brain SPECT

Tc-99m-TRODAT-1 was prepared using a modified version of a previously described method; briefly, Tc-99m-TRODAT-1 was prepared from a freeze-dried kit by adding 1110 MBq of freshly eluted Tc-99m pertechnetate to 5 mL of saline preparation [38]. Tc-99m-TRODAT-1 was obtained over 6 h as a neutral solution (pH = 7.0–7.5) with greater than 90% radiochemical purity, as determined by high-performance liquid chromatography. The shelf life of the lyophilized kit was over two months when stored at room temperature.

Subjects were placed in the supine position and fixed with a head-holder. Three hours after injection of 740 ± 51 MBq (20 mCi ± 1.38) of 99mTc-TRODAT-1, brain SPECT imaging studies were commenced using a dual-headed camera equipped with ultrahigh-resolution fan-beam collimators (Helix SPX, Elscint, Haifa, Israel). Data were acquired in a 128 × 128 matrix at ×1.4 zoom through 360 (180° for each head) rotation at 3° intervals, for 30 s per angle step. Images were reconstructed using a back projection method with a Metz filter. Attenuation correction was performed using the first-order Chang’s method. The trans-axial sections from SPECT were analyzed along the level of the canthomeatal line (CML); this enabled us to obtain coronal and sagittal reconstructions for correct determination of the anatomical regions of the brain.

Image analysis was conducted as described in our previous work [39]. Briefly, to avoid inter- and intra-reader variability in the ROI analysis, the entire task was carried out by an experienced technologist. Regions of interest (ROIs) for the striatum (ST) were drawn manually on overlaid summated SPECT and co-registered MR images for each subject. The test/retest reproducibility had a mean variability of 7.32–9.17% and a correlation coefficient (ICC) of 0.83–0.96 [40]. The SPECT images were analyzed along the level of the canthomeatal line (CML). ROIs were marked for the striatum (ST), caudate nucleus (CA), and putamen (PU) in each hemisphere (with reference to the corresponding MR images) on composite images of the three highest basal ganglia activity slices. The occipital cortices (OC) were also drawn in the same manner and served as background areas. Specific uptake ratios (SURs) were calculated by subtracting the mean counts per pixel in the OC from the mean counts per pixel in the whole ST, PU, or CA regions, and dividing the results by the mean counts per pixel in the background, i.e., (ST-OC)/OC, (PU-OC)/OC or (CA-OC)/OC. The SURs of 99mTc-TRODAT-1 SPECT imaging were compared to values for healthy controls from interpolation and regression methods from our database, some of which have been published [39].

The percent change in SUR before and after HBOT was calculated as:% change = (SURpost − HBOT − SURpre − HBOT)/SURpost − HBOT × 100

#### 2.4.2. Brain MRI

Brain MRI data acquisition and analysis were performed as previously described [27,28]. Brain MR imaging was performed in twenty-one patients within 1 month after CO poisoning (Table 1) on a 1.5 T MR system (Magnetom Vision Plus system; Siemens, Erlangen, Germany) using a single channel circularly polarized head coil. Conventional MR images were obtained using axial T2-weighted fast spin echo [4000/99/1/7 (repetition time (TR)/echo time (TE)/number of excitation/echo train length)], axial FLAIR (9000/110/1; inversion time, 2500), and axial T1-weighted spin-echo (600/14/1) sequences. To minimize inter-subject variation, each ROI was large enough (at least 150 mm^2^), placed at the same size and location in each of the twenty-one patients, and adjusted by re-slicing images using commercial software (AMIRA; Mercury Computer Systems, Chelmsford, MA, USA).

### 2.5. Clinical Diagnosis of Parkinsonism

The assessment for the clinical diagnosis of Parkinsonism was conducted about 1 month after the baseline (pre-HBOT) TC-99m TRODAT-1 brain SPECT imaging studies and repeated 3 months later.

Nineteen patients underwent a neurologic evaluation by an experienced neurologist (S.Y.C.) for parkinsonism; two patients in an acute (critical) condition or who were unable to cooperate did not undergo this evaluation.

The clinical diagnosis of CO-related Parkinsonism was made based on the judgment of symmetric rigidity, bradykinesia, gait disturbances, and postural instability by a movement specialist who was blinded to the clinical diagnosis [41]. The severities of parkinsonian features were evaluated using the Unified Parkinson’s Disease Rating Scale (UPDRS)-part III motor score.

Moreover, measurements of dopamine transport in the striatum of each individual based on TC-99m TRODAT-1 brain SPECT by a trained radiologist were also included in the assessment for the diagnosis of Parkinsonism.

The subjects were classified into two groups based on the three evaluations: Parkinsonism group (n = 13) or non-Parkinsonism group (n = 6).

### 2.6. Statistical Analysis

Linear regression analysis was conducted to assess the relationship between specific uptake in the ST and age. The paired Student’s *t*-test, unpaired *t*-tests, the chi square test, or Fisher’s exact test were performed as appropriate with GraphPad Prism 9 (GraphPad software, La Jolla, CA, USA). A ***p***-value less than 0.05 was considered statistically significant for both the multiple comparisons and correlation analysis. All data are presented as the mean ± SD or 95% CI.

## 3. Results

### 3.1. 99mTC-TRODAT-1 SPECT DAT Availability Correlates with MR Globus Pallidus Lesions

Figure 1 shows example SPECT images of the striatal region of a 35-year-old male subject obtained 3 h post-injection of 99mTc-TRODAT-1 and the MR images before and after HBOT. Besides globus pallidus lesions, T2-weighted FLAIR images showed extensive white matter hyperintensities on both side at baseline (pre-HBOT).

All 21 CO-poisoned subjects in this study developed DNS (21/21). In the baseline (pre-HBOT) 99mTc-TRODAT-1 and MR images, over 70% (15/21) subjects with DNS exhibited necrosis in the globus pallidus on MR images, which was termed GP-positive (GP+). The GP+ patients had significantly lower DAT availability on both sides of the striatum (*** *p* < 0.005) compared to the other patients (GP-negative with higher DAT availability), as shown in Figure 2.

Overall, 68% (13/19) of the subjects with DNS were diagnosed with Parkinsonian syndrome and 32% (6/19) were diagnosed with non-Parkinsonian syndrome; two subjects were not examined for Parkinsonian syndrome. Male patients with DNS had a higher incidence of Parkinsonian syndrome than female patients. Additionally, all male and female GP(+) DNS patients were diagnosed with Parkinsonian syndrome. However, there were no significant differences in the clinical findings between sexes. The MR imaging of the globus pallidus (GP) in patients with Parkinsonism are summarized in Table 2.

### 3.2. 99mTC-TRODAT-1 SPECT DAT Availability Is Significantly Reduced in Parkinsonism Syndrome and HBOT Has Limited Effects

Figure 3 shows example Tc-99m-TRODAT-1 SPECT images of the striatal region of a 33-year-old female subject with non-Parkinsonian syndrome and a 35-year-old male subject with Parkinsonism syndrome at baseline and 3-month follow-up. No significant visible increases in 99mTC-TRODAT-1 SPECT signals were observed in the non-Parkinsonian or Parkinsonian subjects at 3 months after HBOT.

As shown in Figure 4A, the group with Parkinsonian syndrome had significantly lower DAT availability in both striata than the healthy controls (***** p* < 0.001) or patients without Parkinsonian syndrome at baseline (***** p* < 0.001 _PS_ vs. _HC_ or *# p* < 0.05 _PS_ vs. _non-PS_) or 3-month follow-up (*** p* < 0.01 _PS_ vs. _HC_ or *# p* < 0.05 _PS_ vs. _non-PS_). There were no significant differences between the non-Parkinsonian group and healthy controls at baseline or 3-month follow-up. Intriguingly, no significant increase in DAT availability was observed after HBOT in the non-Parkinsonian or Parkinsonian groups.

Figure 4B showed that the SURs on both sides of the striatum increased by 10.46% ± 11.81% post-HBOT in the Parkinsonian group; however, the left striatum presented a significantly higher DAT recovery rate than the right side (Figure 4B, L-ST vs. R-ST, * *p* < 0.05). The data for the healthy controls and for the non-Parkinsonian and Parkinsonian groups before and after HBOT are presented in Table 3.

There were no gender differences in DAT availability at either pre-HBOT or post-HBOT, in agreement with previous reports (Figure 5).

## 4. Discussion

Our results reveal CO-poisoned patients have significantly lower bilateral DAT availability compared to age-matched HC, in agreement with previous Tc-99m-TRODAT-1 SPECT imaging studies [29,42]. We also observed a strong qualitative association between DAT availability (Tc-99m-TRODAT-1 SPECT) and necrosis of the globus pallidus (on MR imaging), and patients with lesions in the globus pallidus had more severe Parkinsonian features and lower DAT availability on both sides of the striatum compared to patients without pallidoreticular lesions. DAT availability in the striatum measured using Tc-99m-TRODAT-1 SPECT was previously reported associated with white matter injures in CO-intoxicated patients [43].

In this study, we demonstrated that the 68.4% (13/19) of DNS subjects who showed Parkinsonian syndrome had significantly lower DAT availability in the bilateral striatum compared to the non-Parkinsonian subjects before HBOT or 3-month follow-up after HBOT. These results are in good agreement with previous findings that CO-poisoned patients had persistently lower bilateral striatal DAT availability, and indications that CO exposure is associated with a higher risk of Parkinsonian syndrome [29,43,44].

In addition to correlation of the structural lesions observed on MR imaging described above, some studies have reported that DAT availability correlates with cognitive function or altered metabolite ratios. For example, Chang et al. (2011) found that patients with lower 99mTc-TRODAT-1 availability in both the caudate and putamen had poorer performance on neuropsychiatric tests and greater atrophy of the thalamus, posterior corpus callosum, cerebral peduncle, and white matter surrounding the globus pallidus compared to patients with higher 99mTc-TRODAT-1 DAT availability [44]. Yang et al. (2021) showed that alterations in DAT availability were associated with altered metabolite ratios of N-acetyl aspartate/creatine, choline-containing compounds/creatine, and myo-inositol/creatine in the left parietal white matter and mid-occipital gray matter (OGM) [42].

The DNS subjects who developed non-Parkinsonian syndrome without pallidoreticular lesions had similar DAT availability before and after HBOT compared to the healthy controls. Therefore, 99mTc-TRODAT-1 striatal DAT availability may not reflect non-Parkinsonian syndrome-related DNS. Alterative treatment should be considered for patients with non-parkinsonian syndrome.

Taken together, the existing evidence indicates bilateral striatal DAT availability coupled with necrosis of the globus pallidus could be an index of the severity of DAT dysfunction or help to screen the risk of Parkinsonian syndrome in patients with CO intoxication, and imaging studies also highlight the clinical application of multiple-modality imaging approaches for the diagnosis or follow-up of CO-intoxicated patients.

To the best of our knowledge, no study has examined the therapeutic effect of HBOT using multi-modality 99mTc-TRODAT-1 SPECT and MR imaging. At the 3-month follow-up (84.6 ± 32.3 days), we found the subjects with Parkinsonian syndrome had significantly lower DAT availability compared to the group with non-Parkinsonian syndrome or healthy controls, although the DAT recovery rate showed a tendency to increase (by ~11%). To date, there have only been a few assessments of recovery of DAT availability in CO-poisoning patients at 6-month [29] or 14-month [45] follow-up. Both of these studies reported no differences in unilateral or bilateral DAT availability, which indicated CO poisoning had long-term effects. These results imply DAT availability recovers very gradually in subjects with Parkinsonian syndrome caused by CO poisoning.

Furthermore, we observed significant higher unilateral recovery rate of DAT in the left striatum than the right striatum in subjects with Parkinsonian syndrome at 3-months after CO poisoning. It should be noted that our finding of unequal recovery in DAT availability on each side of the striatum based on the semi-quantitative SUVs or visual inspection of CO-poisoned patients indicates that the bilateral basal ganglia are asymmetrically affected by CO poisoning, whereas Parkinson’s disease is usually asymmetrical with greater involvement of the putamen [38].

Moreover, CO-associated increases in free radical-mediated or reactive oxygen species (ROS)-mediated dopaminergic neuronal and/or cellular (e.g., erythrocyte) injury is one cause of Parkinson’s disease [4,46,47,48,49]. HBOT accelerates dissociation of CO from hemoglobin, which increases the availability of oxygen to the organs and may prevent DNS [50]. Exposure to mild hyperbaric oxygen activates oxidative metabolism in dopaminergic neurons in the substantia nigra, which inhibits the reduction in dopaminergic neurons and oxidative stress and protects against Bax/Bcl-2-mediated apoptosis, thereby resulting in an attenuation of Parkinson’s disease [51,52].

In addition, CO directly induces cellular changes that provoke immunological or inflammatory damage through several mechanisms [53]. Kuroda et al. reported CO-poisoned patients with DNS had higher myelin basic protein (HBP) levels in cerebrospinal fluid compared to patients without severe symptoms at one month [54].

Thus, the literature and the results of this study imply that CO-induced damage is more closely associated with induction of inflammation and oxidative stress, rather than the direct effects of hypoxia. However, in the current study, there was no change in DAT availability in the patients with Parkinsonian syndrome. Therefore, more investigation is necessary to identify the ideal HBOT regimen to assist the recovery of DAT availability in CO-poisoned patients.

Moreover, it should be noted that the criteria used to evaluate therapeutic efficacy in acute-phase CO poisoning vary between studies and that the protocols employed for HBOT therapy are not consistent. In particular, the profiles of HBOT therapy, including the number of treatments given and the therapeutic pressures, were not consistent across studies from Asia, Europe, and the USA [35,55,56], which may explain why there is no global consensus on effective HBOT regimens for CO poisoning.

Recently, reporting guidelines established by European experts on therapeutic repetitive transcranial magnetic stimulation (rTMS) and other literature have recommended rTMS as a therapeutic strategy for several neurological and psychiatric disorders [57] and regulation of the nodes in the prefrontal cortex involved in the control of fear responses [58]. Multiple high-frequency rTMS sessions over the dorsolateral prefrontal cortex (DLPFC) may improve executive function in patients with PD [59]. However, the efficacy of rTMS in CO-associated Parkinsonian syndrome has not yet been investigated. Thus, it would be interesting to determine whether rTMS would benefit patients with CO-induced Parkinsonian syndrome; specifically, the effects on the action inhibition network (AIN), which processes action control.

### Limitations

The timing of the 99mTc-TRODAT-1 SPECT measurements varied depending on each patient’s emergency status and the availability of the imaging modalities. Therefore, the 99mTc-TRODAT-1 SURs or DAT availability may be affected. Second, the sample size for follow-up MRI or cognitive evaluations was not sufficient enable statistical analysis (data not shown) and may be helpful to identify the correlations with 99mTc-TRODAT-1 DAT availability before and after HBOT.

## 5. Conclusions

Our data demonstrate lower DAT availability was significantly associated with MRI findings of Parkinsonian or non-Parkinsonian syndromes in CO-poisoned patients, even in the acute phase (before HBOT) and after 3-month follow-up. Further, the antioxidant therapy HBOT had limited therapeutic efficacy in terms of recovery of DAT availability at the 3-month follow-up in patients with Parkinsonian syndrome caused by CO-poisoning. This evidence suggests that oxidative stress and neuroinflammation are mainly responsible for the motor control-related damage in CO-induced Parkinsonism, rather than hypoxia.

Moreover, our results also suggest coupled imaging with 99mTc-TRODAT-1 SPECT and MR could represent a useful index to assess the severity of dopaminergic damage, evaluate the risk of developing Parkinsonian syndrome, and evaluate dopaminergic recovery after HBOT in patients with CO poisoning. However, further studies are necessary to determine the efficacy of HBOT and dopaminergic modulation (i.e., rTMS) in patients with CO poisoning-induced Parkinsonian syndrome.

## Figures and Tables

**Figure 1 antioxidants-11-02289-f001:**
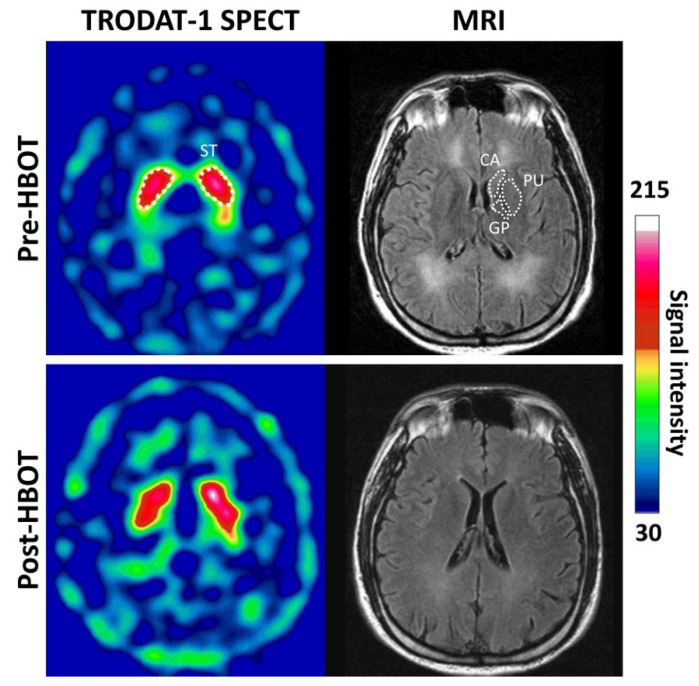
Representative transverse 99mTc-TRODAT-1 DAT SPECT and MR images. The striatal region of a 35-year-old male subject of 99mTc-TRODAT-1 and MR images obtained before and 3 months after HBOT. Regions of interest included the head of the caudate, the putamen and globus pallidus (white dot line) which correspond to 99mTc-TRODAT-1 SPECT. Relative 99mTc-TRODAT-1 DAT uptake in the striatal regions on both sides was low at baseline and slightly increased at 3-month follow-up.

**Figure 2 antioxidants-11-02289-f002:**
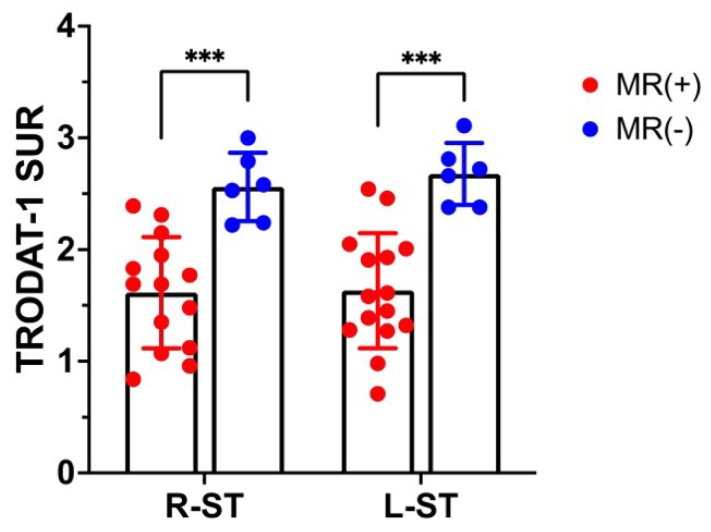
Dopamine transporter (DAT) availability (SURs) and MR findings in the right and left striata. Necrosis of the globus pallidus (GP) on MR images; patients were classified as GP positive (GP+) or GP negative (GP−). The GP+ patients had significantly lower DAT availability on both sides of the striatum compared to GP-patients. Data are mean ± SD; *** *p* < 0.005.

**Figure 3 antioxidants-11-02289-f003:**
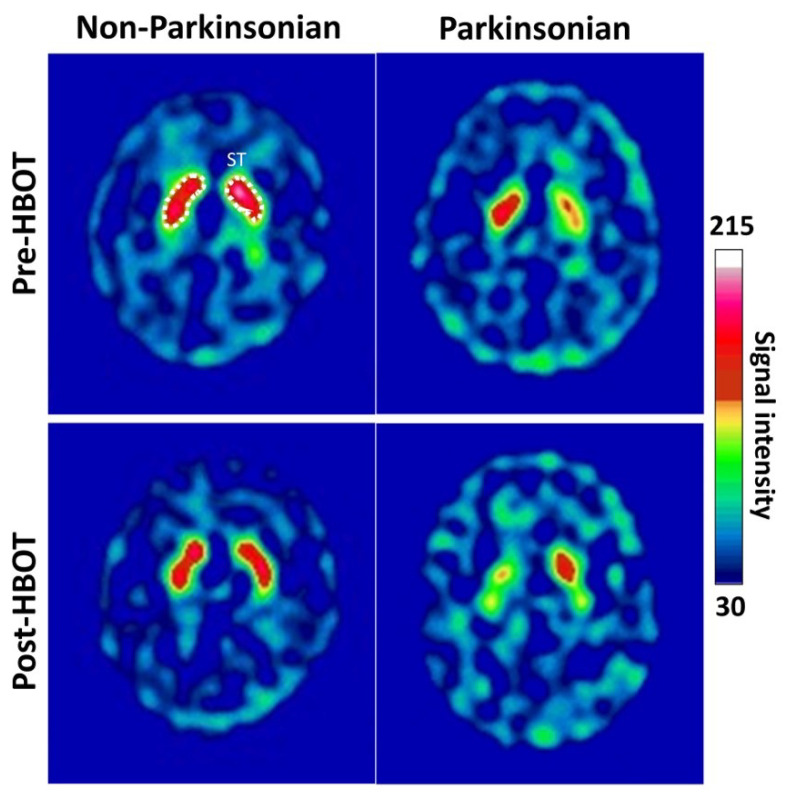
Example transverse 99mTc-TRODAT-1 DAT SPECT and MR images of the striatal regions of patients with non-Parkinsonian and Parkinsonian syndrome before and after HBOT. A 33-year-old female subject with non-parkinsonian syndrome and a 35-year-old male subject with parkinsonism syndrome exhibited similar DAT availability before (baseline) and after HBOT 3-month follow-up, respectively.

**Figure 4 antioxidants-11-02289-f004:**
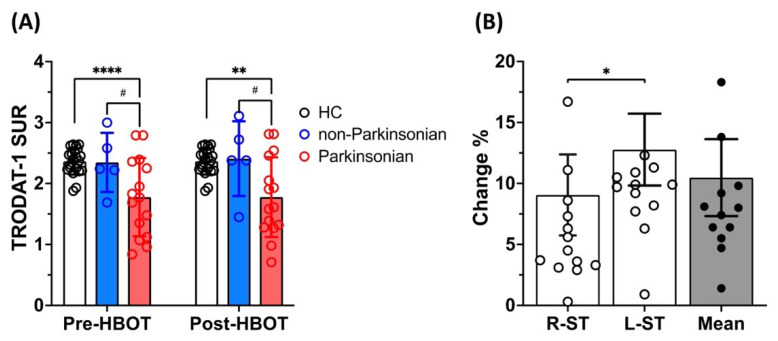
Dopamine transporter (DAT) availability (SURs) for both sides of striatum, as measured by 99mTc-TRODAT-1 SPECT in healthy controls and patients with non-Parkinsonian or Parkinsonian groups before and after HBOT. (**A**) Dopamine transporter (DAT) availability had significant lower in the Parkinsonian group when compared to non-Parkinsonian or health control groups; (**B**) Change (%) in dopamine transporter (DAT) availability (SURs) in the right and left striata (white circles) and in both sides of the striatum (black circles) before and after HBOT. Data are mean ± SD; ** p* < 0.05; *** p* < 0.01; **** *p* < 0.001; # *p* < 0.05 (paired *t*-test).

**Figure 5 antioxidants-11-02289-f005:**
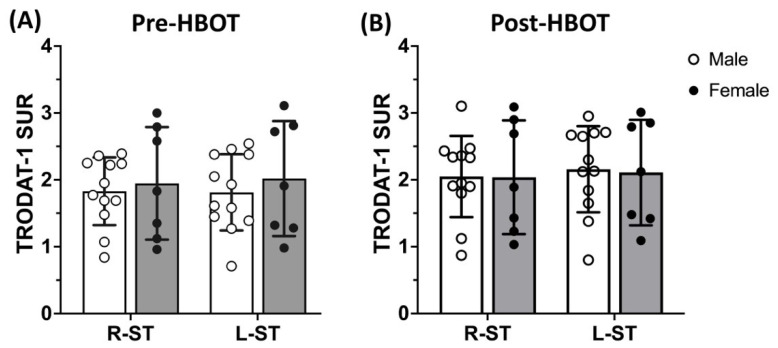
Dopamine transporter (DAT) availability (SURs) in the right and left striata in male and female subjects. There was no difference between male and female subjects at (**A**) baseline (pre-HBOT) or (**B**) 3-month follow up (post-HBOT). Data are mean ± SD. RS, right striatum; LS, left striatum.

**Table 1 antioxidants-11-02289-t001:** Characteristics of the patients with carbon monoxide (CO) poisoning.

	Male	Female	All
Sample size (n)	14	7	21
Age (yr)	38.6 ± 12.5	37.6 ± 11.0	38.6 ± 11.4
Initial COHb (%)	32.6 ± 13.1	28.9 ± 5.3	30.0 ± 10.8
**Before HBOT (baseline)**			
Interval from Dx of CO to MRI (days)	21.9 ± 18.7	16.0 ± 20.2	20.1 ± 21.9
Interval from Dx of CO to TRODAT-1 (days)	34.9 ± 23.3	27.7 ± 24.3	32.3 ± 23.3
**After HBOT (3-month follow-up)**			
Interval from Dx of CO to TRODAT-1 after HBOT (days)	72.3 ± 28.2	106.3 ± 30.0	84.6 ± 32.3

DX: CO-poisoned diagnosed date; CO-carbon monocarboxylate.

**Table 2 antioxidants-11-02289-t002:** Association of MR imaging features in the globus pallidus (GP) with diagnosis of Parkinsonism.

	Male	Female	All
**Clinical diagnosis of DNS**			
DNS Symptom	100% (14/14)	100% (7/7)	100% (21/21)
**MR Findings**			
Incidence of Globus pallidus lesion	78.6% (11/14)	57.1% (4/7)	71.4% (15/21)
**Clinical diagnosis of Parkinsonism**			
Incidence of non-Parkinsonian	25.0% (3/12)	42.9% (3/7)	31.6% (6/19)
Incidence of Parkinsonian	75.0% (9/12) ^a^	57.1% (4/7)	68.4% (13/19) ^a^
*Gender difference*			*NS*
**Parkinsonian correlates with MR findings**			
Parkinsonian with GP (+)	81.8% (9/9)	57.1%% (4/4)	100.0% (13/13)
Parkinsonian with GP (−)	0.0% (0/9)	0.0% (0/4)	0.0% (0/4)
*Gender difference*			*NS*

^a^ Two subjects were not examined for evaluation of Parkinsonian syndrome.

**Table 3 antioxidants-11-02289-t003:** Comparison of the 99mTc-TRODAT-1 DAT SPECT assessments of patients with non-Parkinsonian syndrome (NP) and Parkinsonian syndrome exposed to carbon monoxide poisoning and healthy control (HC) subjects.

	HC	Non-Parkinsonian (NP)						Parkinsonian							
Sample Size (n)	21	6						13							
**DAT availability**		Pre-HBOT	*p* vs. HC	Post-HBOT	*p* vs. HC	*p* vs. pre-HBOT	Change %	Pre-HBOT	*p* vs. HC	*p* vs. non-PS	Post-HBOT	*p* vs. HC	*p* vs. pre-HBOT	*p* vs. non-PS	Change %
Both sides of striatum	2.36 ± 0.22	2.38 ± 0.55	NS	2.51 ± 0.49	NS	NS	5.74% ± 5.74%	1.70 ± 0.60	****	#	1.92 ± 0.65	**	NS	#	10.46% ± 11.81%
Right striatum		2.35 ± 0.48	---	2.45 ± 0.48	---	NS	4.56% ± 1.39%	1.70 ± 0.60	---	#	1.90 ± 0.70	---	NS	0.0603	8.29% ± 13.30%
Left striatum		2.41 ± 0.61	---	2.57 ± 0.53	---	NS	6.92% ± 6.56%	1.70 ± 0.61	---	#	1.98 ± 0.67	---	NS	#	12.64% ± 11.18%

Data are mean ± SD; NS: no significant difference; ---: not available; ** *p* < 0.05, **** *p* < 0.001 compared to HC group; # *p* < 0.05 compared to patients with non-parkinsonian syndrome.

## Data Availability

The data presented in the study are available in the article.

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
