# Peer review of "Multi-Modal Synergistic 99mTc-TRODAT-1 SPECT and MRI for Evaluation of the Efficacy of Hyperbaric Oxygen Therapy in CO-Induced Delayed Parkinsonian and Non-Parkinsonian Syndromes"

_antioxidants, 2022, doi:10.3390/antiox11112289_

Round 1
Reviewer 1 Report
This interesting study describes the usefulness of functional-99mTc-TRODAT-1 SPECT assessment of dopamine transporter availability and structural MR imaging of necrosis in the globus pallidus to evaluate the risk of developing the parkinsonian syndrome. Most of the data are of confirmative nature, but the imaging approach is rather novel.
There are some specific comments.
Page 2:
»Neuroanatomical abnormalities following CO poisoning are detected on MRI in the globus pallidus, and symmetrical bilateral basal ganglia«. GLOBUS PALLIDUS IS A PART OF THE BASAL GANGLIA. WHAT ELSE OF THE BASAL GANGLIA IS MEANT BY THIS STATEMENT?
Page 3:
»None of the subjects had a current or past history of neuropsychiatric disorders or a family history of movement disorders...« HOW DID YOU ASSESS THE»ABSENCE OF NEUROPSYCHIATRIC DISORDERS« IN PATIENTS ATTEMPTING A SUICIDE? NO DEPRESSION OR ANYTHING ELSE?
Page 4, Table 1:
What is »Intravel«? Maybe Interval?
Page 5:
Single channel head coil? How much time did it take to acquire the images?
What was the use of T2, T1 and FLAIR – not described, data not presented and not shown on figures.
Page 6:
Fig.1: It is clear that other parts of the brain are affected as well (judging from images). Why did you not comment on this MRI finding?
This interesting study describes the usefulness of functional-99mTc-TRODAT-1 SPECT assessment of dopamine transporter availability and structural MR imaging of necrosis in the globus pallidus to evaluate the risk of developing the parkinsonian syndrome. Most of the data are of confirmative nature, but the imaging approach is rather novel.
There are some specific comments.
Page 2:
»Neuroanatomical abnormalities following CO poisoning are detected on MRI in the globus pallidus, and symmetrical bilateral basal ganglia«. GLOBUS PALLIDUS IS A PART OF THE BASAL GANGLIA. WHAT ELSE OF THE BASAL GANGLIA IS MEANT BY THIS STATEMENT?
Page 3:
»None of the subjects had a current or past history of neuropsychiatric disorders or a family history of movement disorders...« HOW DID YOU ASSESS THE»ABSENCE OF NEUROPSYCHIATRIC DISORDERS« IN PATIENTS ATTEMPTING A SUICIDE? NO DEPRESSION OR ANYTHING ELSE?
Page 4, Table 1:
What is »Intravel«? Maybe Interval?
Page 5:
Single channel head coil? How much time did it take to acquire the images?
What was the use of T2, T1 and FLAIR – not described, data not presented and not shown on figures.
Page 6:
Fig.1: It is clear that other parts of the brain are affected as well (judging from images). Why did you not comment on this MRI finding?
One of my concerns is missing control group (but this is a limitation of the study).
Reviewer 2 Report
Yeh and colleagues in the present research article entitled ‘Multi‐modal synergistic 99mTc-TRODAT-1 SPECT and MRI for evaluation of the efficacy of hyperbaric oxygen therapy in CO-induced delayed Parkinsonian and non-Parkinsonian syndromes’ is a well-written and useful summary of the status of knowledge on the evaluation of 99mTC-TRODAT-1 SPECT to assess the severity of DAT damage associated with MRI findings after CO poisoning. For this purpose, authors decided to assess DAT availability before and after HBOT using 99mTC-TRODAT-1 SPECT, and to evaluate multi‐modal synergistic 99mTC-TRODAT-1 SPECT and MRI as an index for parkinsonian or non-parkinsonian syndromes in patients with CO poisoning.
The main strength of this manuscript is that it addresses an interesting and timely question, describing how coupled imaging with 99mTc-TRODAT-1 SPECT and MR could represent a useful index to assess the severity of loss of dopamine transporter (DAT), evaluate the risk of developing parkinsonian syndrome, and DAT recovery in patients with CO poisoning. In general, I think the idea of this review is really interesting and the authors’ fascinating observations on this timely topic may be of interest to the readers of Antioxidants. However, some comments, as well as some crucial evidence that should be included to support the author’s argumentation, needed to be addressed to improve the quality of the manuscript, its adequacy, and its readability prior to the publication in the present form, in particular reshaping parts of the Introduction and Discussion sections by adding more evidence and theoretical constructs.
Please consider the following comments:
· Abstract: According to the Journal’s guidelines, the abstract should be a single paragraph without headings, and should be a total of about 200 words maximum. Please correct the actual one.
· A graphical abstract is highly recommended.
· Introduction: I suggest the authors to reorganize the Introduction section, which seems inhomogeneous and dispersive. I think that more information pathophysiology and symptoms of Parkinson’s disease (PD) would provide suitable background here. Thus, I suggest the authors to make an effort to provide a brief overview of the pertinent published literature that offer a perspective on neurobiological alterations that lead to this neurodegenerative disorder, because as it stands, this information is not highlighted in the text and makes the reader unable to understand the role of dopamine neuronal damage in producing CO-induced parkinsonism. Authors should briefly describe the pathogenesis of this inflammatory-mediated dopaminergic degeneration, specifically focusing on describing the significant loss of dopamine characterizing fronto-temporal degeneration in PD disrupting voluntary motor control and causing the characteristic symptoms of PD: a recent review examined pathophysiological basis and biomarkers of PD pathology and investigated molecular signs of neuroinflammation in neurodegenerative diseases (doi:10.1016/j.parkreldis.2017.07.033), while recent evidence from studies conducted on healthy individuals focusing on action control supported a causal role of dopamine in action control, addressing how PD is accompanied by impairments in covert cognitive processes (https://doi.org/10.3389/fnbeh.2022.946263) underlying goal-directed motor functioning (e.g., action planning, conflict adaptation, motor inhibition) (https://doi.org/10.3389/fnbeh.2022.998714; https://doi.org/10.1101/cshperspect.a009282), and that dopaminergic medication may modulate these action control components.
· Patient enrollment: Data about participants and information about clinical assessment for patients’ selection are not adequately explained. For this reason, I would ask the authors to specify inclusion criteria for patients involved in this study, like severity of disorder. Also, could the authors specify how did they estimate the exact number of participants? Did they use a power analysis?
· Clinical diagnosis of parkinsonism: Please provide more information about the clinical tests that were used in the diagnosis of Parkinson’s disease, and how they were administered.
· Results: In my opinion, this section is well organized, but it illustrates findings in an excessively broad way, without really providing full statistical details, to ensure in-depth understanding and replicability of the findings. I suggest rewriting this section more accurately, and to present statistical data in tables.
· In my opinion, I think the ‘Conclusions’ paragraph would benefit from some thoughtful as well as in-depth considerations by the authors, because as it stands, it is very descriptive but not enough theoretical as a discussion should be. Authors should make an effort, trying to explain the theoretical implication as well as the translational application of their research.
· Figures and Tables: According to the Journal’s guidelines, please add an explanatory caption for each figure/table within the text. Also, please change the scale of the vertical axis and use the same minimum/maximum scale value in all the figures’ graphs. I also suggest to modify all figures for clarity and provide higher-quality images because, as it stands, the readers may have difficulty comprehending them. In my opinion, data settings are overcrowded and written with a very small font. I suggest to better organize the graphs’ space in all the figures, to provide a better understanding and a direct interpretation of the results.
· References: Authors should consider revising the bibliography, as there are several incorrect citations. Indeed, according to the Journal’s guidelines, they should provide the abbreviated journal name in italics, the year of publication in bold, the volume number in italics for all the references.
Overall, the manuscript contains 2 tables, 5 figures and 42 references. I believe that the manuscript might carry important value describing how coupled imaging with 99mTc-TRODAT-1 SPECT and MR could represent a useful index to assess the severity of loss of dopamine transporter (DAT), evaluate the risk of developing parkinsonian syndrome, and DAT recovery in patients with CO poisoning.
I hope that, after these careful revisions, this paper can meet the Journal’s high standards for publication.
I am available for a new round of revision of this paper. I declare no conflict of interest regarding this paper.
Best regards,
Reviewer
Round 2
Reviewer 2 Report
The authors did an excellent job clarifying all the questions I have raised in my previous round of review. Currently, this paper entitled ‘Multi‐modal synergistic 99mTc-TRODAT-1 SPECT and MRI for evaluation of the efficacy of hyperbaric oxygen therapy in 3 CO-induced delayed Parkinsonian and non-Parkinsonian syndromes’, is a well-written, timely piece of research that examined how coupled imaging with 99mTc-TRODAT-1 SPECT and MR could represents a useful index to assess the severity of loss of dopamine transporter (DAT), evaluate the risk of developing parkinsonian syndrome, and DAT recovery in patients with CO poisoning.
Overall, this is a timely and needed work. It is well researched and nicely written, and describes in detail the use of 99mTC-TRODAT-1 SPECT to assess the severity of DAT damage associated with MRI findings after CO poisoning.
I believe that this paper does not need a further revision, therefore the manuscript meets the Journal’s high standards for publication.
I am always available for other reviews of such interesting and important articles.
Thank You for your work.